# Sea-Surface Target Visual Tracking with a Multi-Camera Cooperation Approach

**DOI:** 10.3390/s22020693

**Published:** 2022-01-17

**Authors:** Jinjun Rao, Kai Xu, Jinbo Chen, Jingtao Lei, Zhen Zhang, Qiuyu Zhang, Wojciech Giernacki, Mei Liu

**Affiliations:** 1Shanghai Key Laboratory of Intelligent Manufacturing and Robotics, School of Mechatronic Engineering and Automation, Shanghai University, Shanghai 200444, China; jjrao@shu.edu.cn (J.R.); xkai@shu.edu.cn (K.X.); jbchen@shu.edu.cn (J.C.); jtlei@shu.edu.cn (J.L.); zhangzhen_ta@shu.edu.cn (Z.Z.); zhang-qiuyu@shu.edu.cn (Q.Z.); 2Institute of Robotics and Machine Intelligence, Faculty of Control, Robotics and Electrical Engineering, Poznan University of Technology, Piotrowo 3a, 60-965 Poznan, Poland; wojciech.giernacki@put.poznan.pl

**Keywords:** multi-camera cooperation, sea-surface moving target, visual tracking system, visual detection

## Abstract

Cameras are widely used in the detection and tracking of moving targets. Compared to target visual tracking using a single camera, cooperative tracking based on multiple cameras has advantages including wider visual field, higher tracking reliability, higher precision of target positioning and higher possibility of multiple-target visual tracking. With vast ocean and sea surfaces, it is a challenge using multiple cameras to work together to achieve specific target tracking and detection, and it will have a wide range of application prospects. According to the characteristics of sea-surface moving targets and visual images, this study proposed and designed a sea-surface moving-target visual detection and tracking system with a multi-camera cooperation approach. In the system, the technologies of moving target detection, tracking, and matching are studied, and the strategy to coordinate multi-camera cooperation is proposed. The comprehensive experiments of cooperative sea-surface moving-target visual tracking show that the method used in this study has improved performance compared with contrapositive methods, and the proposed system can meet the needs of multi-camera cooperative visual tracking of moving targets on the sea surface.

## 1. Introduction

In recent years, with the continuous progress of computer and image processing technology, computer vision has been widely used in intelligent monitoring, visual navigation, industrial vision systems, and other fields in which visual detection and tracking are particularly valued by researchers [1]. Visual detection and tracking are to extract and analyze the key information in the video stream through various image-processing algorithms, and then react according to the analysis results, so as to achieve target segmentation and tracking. Visual tracking systems have broad application prospects in the field of artificial intelligence such as multi-robot cooperation, video surveillance, video-based human–computer interaction, and unmanned driving.

This paper mainly studies the tracking and detection of moving targets on the sea surface. Due to the maneuvering characteristics of sea-surface moving equipment and the influence of the marine environment, compared with land-target tracking and detection, the visual signal processing of sea moving targets has the following difficulties: (a) The relative position and attitude of the tracked target changes abruptly, which will lead to blur or dance of the video image, and the difference between two adjacent frames of the image becomes too large, or even deviate from the target field of view; (b) The tracked target appears and disappears between the waves, so the size and shape of the captured target will change frequently, which increases the difficulty of dynamic image matching; (c) Each camera installed on a vehicle moves dynamically and rolls back and forth heavily, which will affect the public viewing area of each camera in real time [2]. If there is no coordination between the cameras, there will be a big error in target tracking.

To solve the above difficulties, this paper studies the related technologies based on multi-dimension visual information, and a multi-camera cooperative visual tracking system for sea-surface moving targets is designed. The system can be deployed on a variety of mobile carriers, such as unmanned surface vehicles (USVs), and then realize the autonomous tracking and monitoring of sea-surface targets by multi-USV [3].

### 1.1. Related Works

At present, cameras implemented in different surveillance systems generally work independently of each other. In this tracking mode, the viewing angle is limited and the dynamic adaptability is poor, so it is difficult to achieve expected accuracy and reliability. In the scenario of target tracking using multiple cameras, if the image and pose information of cameras could be shared with each other. In other words, if cooperation among cameras is realized, the scope of video surveillance can be expanded and sustainable ability of target tracking can be improved. Moreover, the target is monitored from different observation directions of multiple cameras, which can effectively avoid the target being blocked or lost from the observation scope of a single camera [4,5,6].

Compared with single cameras, target detection and tracking with multi-camera cooperation has unique advantages in many aspects. Therefore, related technologies have attracted more and more researchers’ interest [7]. Khan SM and Shah M studied a method of multiple points of view, where all cameras obtain a sequence of images collected in a collaborative framework, and at the same time multiple prospects of likelihood information fuse together, using two-dimensional structure positioning to track a moving object in the scene, to solve the problem of shade and improve robustness [8]. Yan WL proposed a tracking algorithm based on an interactive multi-model particle filter for moving targets in the sky environment on the basis of dual-camera co-tracking [9]. Ighrayene M et al. proposed a Bayesian tracking matching and coordination strategy based on binary robust invariant scalable Key point (BRISK) based on the existing features, search mechanism, and generalization of target representation to solve obstacles such as occlusion and transience that are often encountered in target tracking [10]. In this method, a Bayesian tracker is used as the main tracker and a fast-matching method is adopted to retrieve the lost target. Zheng Y et al. proposed using topology structure to allocate multiple cameras for moving-target tracking [11]. Wang T et al. proposed a collaborative relay tracking algorithm based on multiple cameras, which used a genetic algorithm to shorten the total moving distance of cameras in the tracking process [12]. Zhang GF et al. studied a multi-camera visual-positioning technology, which combined KNN-based background segmentation algorithm and a KCF algorithm with a feature-matching algorithm to achieve continuous and reliable target tracking. The technology can obtain real-time position of personnel in a fixed scene [13]. To solve the problem of poor robustness caused by occlusion and fast movement in multi-target tracking, Yao SS et al. proposed an opt-based attitude correlation algorithm, where local attitude matching is used to solve the occlusion problem and optical flow is used to reduce the influence caused by fast motion [14]. Multiple cameras can be used to track multiple different targets at the same time and the target trajectory under multiple cameras can be generated automatically. Neehar P et al. proposed a multi-camera vehicle tracking and re-identification system [15]. The system introduces an unsupervised excitation layer to enhance learning to solve the task of vehicle recognition, and uses re-identification feature extraction to calculate the distance matrix and obtain the multi-camera vehicle track after clustering. The method achieves the best results on the vehicle identification dataset Vehicle-ID. In 2021, Srigrarom et al. proposed a real-time multi-camera system. The system can realize target recognition by fusing the target trajectory and relative position information of the target between each camera. It can simultaneously track and locate multiple targets in three-dimensional space [16]. Although positive results are reported, sea-surface targets are not the subjects in this research. Therefore, what motivates us is to develop a novel visual tracking system for sea-surface targets with a multi-camera cooperation approach.

### 1.2. Multi-Camera Cooperative Moving-Target Visual Tracking System

The moving- target visual tracking (MTVT) system with multi-camera cooperation is proposed as shown in Figure 1. This system consists of multiple target-tracking units and a human–machine interface (HMI). Each unit is composed of a pan–tilt camera and a small-sized industrial computer with a wireless communication module. Every unit is able to track an allocated target independently while exchanging image and coordination information through a communication module. The HMI gathers and fuses information from every unit, and coordinates all units to track the target cooperatively through the specified strategy. In the whole system, only a single unit acts as the master tracking unit (MTU), and other units in the system assist the MTU. When the MTU loses the target, the MTU will be re-selected. Since the camera is the basic component of each unit, we also use ‘camera’ to refer to the target-tracking unit.

In order to overcome the problems of sea-surface application and improve the target-tracking capability of each unit, technologies for target detection, matching, and tracking are studied. Regarding target detection, an improved target-detection algorithm based on the Three Frame Difference method (TFD) and Mixed Gaussian Background modeling (MGBM) method is proposed to obtain a more complete moving-target contour and more accurate position. As for target tracking, an improved target-tracking method based on a multi-domain convolutional neural network (MDNet) is researched to improve stability and robustness of target tracking. Meanwhile, in the motion-control process, each camera’s pan-and-tilt attitude is controlled to lock and track the moving target synchronously. As for target matching, an improved target-matching method based on a Speeded Up Robust Features (SURF) algorithm and RANdom SAmpling Consensus (RANSAC) algorithm is implemented to reduce the false-matching rate and improve the matching effect.

To coordinate all tracking units to work together, coordination strategies are researched, including pan-and-tilt control, MTU selection, the unit behavior state model, and cooperative tracking strategy.

This paper is organized as follows. The technologies of target detection, tracking, and matching methods are studied in Section 2. The coordination strategy are introduced in Section 3. Experimental results are shown and analyzed in Section 4. Finally, conclusions and prospects are given in Section 5.

## 2. Technologies of Moving-Target Visual Detection, Tracking, and Matching

### 2.1. Moving-Target-Detection Algorithm Based on Mixed Gaussian Background Modeling and Three-Frame Difference Method

Target-detection technology is the basis of multi-camera cooperative target tracking, and the accuracy of target detection directly affects the matching and tracking of the same target between multiple cameras.

In this paper, the background difference method and background modeling method are mainly referenced as moving-target-detection methods. In the background difference method, the difference in image between the current frame and the corresponding background model is calculated, and foreground region in the image can be obtained. This algorithm is mainly applicable to known background. TFD is a typical background difference method. In TFD, two differential images can be obtained using three successive frames, and logical “and” operation is performed on the differential images to extract a complete moving-target contour [17,18]. However, in this algorithm, hollow phenomena easily appear when detecting moving targets, resulting in target-information loss. Background modeling converts the problem of moving-target detection in video into a binary classification problem according to the current background estimation [19,20]. MGBM is a multi-modal model of the background modeling method which is based on a weighted average of multiple Gaussian probability density functions to achieve smooth approximation of the density distribution function under any state, so that it can adapt to changes of each point in dynamic background and integrate interference into the background. However, the moving-target image extracted in this method has few edge pixels and lacks contour information of the target.

A camera in motion will cause new problems. According to whether the camera is in motion, moving-target visual-detection problems can be divided into two categories: one is target detection, where the camera is stationary and most of the scene in the camera field is stationary; the other is target detection when the camera is in motion, which will cause background change in its video frame [21,22,23]. In this study, we consider the applications where the cameras are installed on movable vehicles and the background changes dynamically. In this condition, the foreground segment and the detection results obtained by using a single, traditional, moving-target-detection algorithm are not satisfactory.

In order to combine the advantages of MGBM and TFD, an improved moving-target visual-detection algorithm is proposed and used to improve accuracy in the sea-surface environment. The flowchart of the algorithm is shown in Figure 2.

The algorithm mainly consists of three steps:(1).The acquired image sequences are preprocessed frame by frame, including graying, filtering, and global-motion compensation. The purpose of this step is to weaken noise, enhance image details, and improve efficiency of effective information extraction.(2).MGBM and TFD are used for target detection, respectively.(3).The processed image is binarized, and the mass center of the moving target processed by MGBM is used as the center, and then the logical “and” operation is carried out with the foreground region extracted by TFD. Then, image shape is processed and the final moving-target image is obtained.

In step (3), the process of obtaining the mass center of the moving foreground is as follows: the appropriate threshold is obtained by the Otsu algorithm; the target contour is retrieved; finally, the mass center is calculated through the gray centroid method, in which the gray value of each pixel in the image is regarded as the “mass” of that point, and the center of the region can be calculated as follows:(1)$$\bar{u}=\frac{\sum_{\left(u,v\right)\in \Omega }u\times f\left(u,v\right)}{\sum_{\left(u,v\right)\in \Omega }f\left(u,v\right)}$$
(2)$$\bar{v}=\frac{\sum_{\left(u,v\right)\in \Omega }v\times f\left(u,v\right)}{\sum_{\left(u,v\right)\in \Omega }f\left(u,v\right)}$$
where *u* and *v* are pixel coordinate values and f(u, v) is the gray value of a pixel with the given coordinates.

### 2.2. Moving-Target-Tracking Method Based on MDNet

The essence of moving-target tracking is to find out the position of a target in an image according to characteristics of texture, color, brightness, and so on [24,25]. The target-tracking performance with a single camera is the basis of a multi-camera cooperative tracking system. In recent years, many scholars have tried to integrate neural networks into target-tracking algorithm, hoping to improve long-term stability and the ability to adapt to environment and target changes [26,27]. Among them, Deep Learning Trackers (DLT) integrate Deep Learning into a tracking algorithm for single target [28]. After that, new methods have emerged and used for target features extraction, online tracking, and so on [29,30].

A convolutional neural network (CNN) has great advantages in representing visual data compared with traditional model-based and feature-based tracking algorithms, and they have been widely used in various computer vision tasks, such as image classification, semantic segmentation, and so on [31,32,33,34]. However, it is not easily applicable in visual tracking, since it is difficult to obtain useful datasets including diverse combination of targets and backgrounds with different appearance, and motion modes of different categories of targets in different video sequences. To make CNN adapt to inconsistency of targets between video sequences, Nam H and Han B proposed a multi-domain network (MDNet) architecture [35]. They regard each video as a separate domain, and the multi-domain network learns commonness of a target from multiple annotated video sequences, so that a target can be tracked in each video. Compared with some new algorithms, such as the Siamese regional proposal subnetwork (SiameseRPN) [36], accurate tracking by overlap maximization (ATOM) [37], and discriminant model prediction (DiMP) for tracking [38], MDNet is not the most outstanding in every aspect of tracking performance, however, its tracking accuracy is satisfactory, and it also has good results in application of target tracking.

The architecture of MDNet used in this paper is shown in the purple dashed box in Figure 3. The MDNet has five shared layers, including three convolution layers and two full-connection layers. In convolution layers Conv 1–3, features are extracted adaptively with VGG-M network structure. Fully connected layers FC 4–6 are used to abstract feature-map representation into a one-dimensional vector. Full-connection layer FC6 will contain multiple branches. If there are *K* training video sequences, FC6 at the end of the network has *K* branches correspondingly. Every branch of FC6 is actually a binary classification layer. The network parameters of shared layers in MDNet are updated during training using all videos, while the parameters of full-connection layer FC6 iteratively update only during training with its corresponding video sequence. Finally, MDNet outputs the confidence of the target and background of the image.

The MDNet should be pretrained before it is used for tracking. In this study, the pretraining process included two stages: training of the first frame and training of the subsequent frames, as shown in Figure 3.

For the first-frame image with a moving target, the boundary box of the target will be initiated using the target-detection algorithm in Section 2.1 firstly. Secondly, based on the initial bounding box, 1000 positive samples, whose intersection over union (IoU) are not less than 0.6, are generated and input into MDNet, so the feature graph representation after convolutional layers processing can be obtained. Thirdly, after dimensional transformation, the obtained features are all used to train the bounding box regression model together with the true boundary box of the target. Meanwhile, the positive and negative samples of the first frame and features obtained by three convolutional layers are extracted and retained. To fine-tune full-connection layers, the network is trained 30 times iteratively. In every iteration, 32 positive samples and 1024 negative samples of convolution layer features are selected randomly and used as inputs to the training network, and their scores and forward propagation losses are calculated, respectively.

For the subsequent frame training, 256 candidate samples based on the target bounding box of the previous frame are generated as an input of the network, and the score of each candidate is calculated. Then, the average region of the top five samples is taken as the target bounding box of the current frame. If the average score of the top five is bigger than the threshold, it is determined that the network tracked the target, and then these steps will be followed: (1) generate 50 positive samples with IoU no less than 0.7 according to the currently predicted target box; (2) save the features of the convolution layers by processing these positive samples; (3) update the positive and negative samples of the latest 20 frames. Otherwise, the target box of the previous frame will be copied as the target-tracking result of the current frame while expanding area and researching once, and then positive and negative samples of the latest 20 frames will be used to train the network iteratively.

When MDNet is used for tracking targets online, a new network is constructed by combining shared layers in pretrained networks with a new, online, updating binary classification layer FC6. Online tracking will be performed by evaluating a candidate box randomly sampled around the target image in the previous frame. Meanwhile, robustness and adaptability of network are updated according to long-term and short-term appearance changes of the target, respectively.

During online tracking, *N* candidate boxes around the last frame, $x^{1}$,……, $x^{N}$, are selected randomly and used to evaluate the target probability $f^{+}\left(x^{i}\right)$ or background probability $f^{-}\left(x^{i}\right)$, select the one with the highest $f^{+}\left(x^{i}\right)$ value as the best candidate of target, and the corresponding $x^{*}$ is regarded as the target box of the current frame.
(3)$$x^{*}=\underset{x^{i}}{\;arg\;max}f^{+}\left(x^{i}\right)$$

In addition, since a visual platform is composed of multiple units, when one unit detects a moving target, the video frame with target information of the camera is pre-trained, and the pre-training result is applied to the multi-domain convolutional neural network of other units for subsequent tracking.

### 2.3. Moving-Target-Matching Method for a Multi-Camera System

As we know, even for the same target in the same scene, color and shape of the target and position in the image field will be different for each camera from a different view angle. Therefore, the premise of cooperative tracking is to match the same moving target in field of view of multiple cameras. The purpose of target matching is to judge whether the tracked target in each camera is the same target, so as to lock the common moving target for multi-camera cooperative tracking.

Among many target-matching algorithms, the Speeded Up Robust Features (SURF) algorithm optimizes feature point extraction and feature vector description based on a Scale Invariant Feature Transform (SIFT) algorithm, which not only maintains the good performance of the SIFT algorithm, but also improves the calculation efficiency [39,40,41].

However, due to highly dynamic changes in sea-surface targets and moving view angles of multiple cameras, there will be many mismatching points in target matching using only a SURF algorithm. As a data-screening method, the RANdom SAmpling Consensus (RANSAC) algorithm has good performance when screening out mismatched points-pairs [42]. In RANSAC, it is assumed that the input data include normal data and abnormal data. Normal data can be described by a mathematical model with some parameters which can be obtained by iteratively selecting a group of random subsets from the data. Abnormal data cannot be described by a model, and their value is not within the range of correct data.

In order to match the true feature points of moving targets detected by different cameras, an improved target-matching method combining SURF and RANSAC algorithms is proposed in this paper, as shown in Figure 4. In this method, the RANSAC algorithm is applied to check the feature points-pairs and descriptors obtained by the SURF algorithm, and the regions with the largest number of matching pairs are regarded as of the moving target.

## 3. Coordination Strategies

As mentioned in Figure 1, the coordinate controller is a key part of the cooperative MTVT system, in which coordination strategy plays a decisive role. Coordination strategy includes strategy of selection of the MTU, the behavior state alternation model of cameras, and cooperative tracking strategy of considering complex perspective relationship.

### 3.1. Pan-and-Tilt Control

Pan-and-tilt (PT) control is necessary for moving-target visual tracking. In order to control cameras tracking a moving target smoothly, both tracking speed and position are taken into consideration. The rotation speed of a camera has eight levels from slowest speed 1 to fastest speed 8. The view field or the picture of camera is divided to nine regions by a central circle and range of angles, as shown in Figure 5. According to the coordinate system, the circle has a center *C*(*W*/2, *H*/2) and radium *R*. If the center of the target image is *T(x,y)*, then the vector r can be obtained by:(4)$$r\;=\;\left(x\;-\;W/2\right)\hat{i}\;+\left(y\;-\;H/2\right)\hat{j}$$

Therefore, the PT direction of the camera could be given as shown in Table 1. It can be seen that start- and stop-rotation depend on the relationship between vector ***r*** and radius *R* of the center area of the camera field of view. Furthermore, the problem of solving start and end time of the PT is transferred to calculation of rotation-direction parameters. The PT speed-control strategy of the target-tracking system studied in this section is as follows:(1).Initialize partition of camera view field and set PT rotation speed to 1.(2).Adaptively adjust control parameters of PT. If the target is not in the central partition of the camera, rotate camera PT according to Table 1. Meanwhile, the position offset of target in two adjacent frames is calculated. According to the obtained offset, the rotation speed of PT is adjusted to keep the motion of the camera and target as consistent as possible, and to keep the target always placed in the center of the field of view.(3).Correct PT rotation parameters dynamically according to the position offset of the target in two adjacent frames. Calculate ***r*** frame by frame. If |***r***| > R, maintain rotation of PT, and adjust rotation speed in real time according to the rotation result of the next frame until the moving target can be tracked smoothly.

### 3.2. The MTU Selection

In our MTVT system, each tracking unit can track targets not only independently, but also cooperatively through sharing information [43,44]. There is one MTU in the system, and the other cameras act as slave tracking cameras (STCs). The target being tracked by the MTU is regarded as the current common tracked target for each STC. The MTU is selected through following policies:(1).The tracking unit that captures the moving target first is selected as the MTU.(2).If two or more cameras detect the moving target at the same time, the camera whose current target image is more centered in the window, and target size in the picture is larger than the threshold is chosen, as the MTU.(3).If the detected targets are all near the center of image of multiple cameras, the camera with a larger size of the target image is chosen as the MTU.(4).When the center of gravity of the target is out of the view field of the MTU, or half of the target image is outside the MTU view window, or the MTU loses the target, selection of the MTU from the others cameras occurs according to (1)–(3).

### 3.3. Unit Behavior-State Model

In order to coordinate multiple cameras for moving-target tracking, a behavior state model based on a finite state machine is designed. In this model, any tracking unit at any time will be in one of following four states as shown in Figure 6:(1).Waiting state. When a tracking unit loses communication with the host controller or computer, the unit will be in this state. Once communication is set up, the unit will change to detecting state.(2).Detecting state. In this state, the target is out of the range of view of the unit, so the target will be detected by using shared information from other units, meanwhile adjusting the PT of the camera under control of the host. Once the target is detected and matched, the unit will change to tracking state.(3).Tracking status. In this state, the moving target will be tracked continuously by adjusting the PT of the camera using the control strategy in Section 3.1. If the target is out of lock temporarily, while the target is still in the field of view of camera, the unit will change to the state of out of lock. If the target is out of viewing field, the unit will switch to detecting state.(4).Out-of-lock state. In this state, the target is out of lock, the camera will stop rotating and begin to search for a moving target in the field of view. If the target is relocked, the unit will revert to tracking state, otherwise switching to detecting state.

The state information of each unit will be shared with each other to guide all units to track the moving target.

### 3.4. Cooperative Tracking Strategy

In our MTVT system, each unit has an independent view field which is related to position and viewing angle of the unit, so combinations of the view fields of all units are complex, and the view relationship among cameras will have an impact on target-tracking effects and target positioning. For our system, there are three typical pose relations among cameras and targets as shown in Figure 7, and coordination strategies are proposed as follows:(1).Pose relation of tracking beginning. As shown in Figure 7a, the system begins to detect the target, and the target is tracked by no more than one unit. The first unit finding the target acts as the MTU. Other units in detecting state control and rotate PT to detect and match the target from the view field of the MTU to save detecting time.(2).Pose relation of partial cooperation. As shown in Figure 7b, there are at least two units tracking the target, and they are in partial-cooperation pose relation, and the axes of view of cameras have at least one intersection point near the target. By calculating and sharing the position information of these intersection points, the units in detecting state or in out-of-lock state will be guided to find the target more quickly.(3).Pose relation of full cooperation. All units cooperate to track the target and their view field is superimposed. In this pose relation, the position of the target could be estimated by calculating the average value of the intersection points of view axes of cameras, as green points and circles show in Figure 7c.

By using the above strategy and analyzing the pose layout and the state of the cameras in tracking process, the target can be tracked continuously even if some cameras fail to track the target temporarily, and the unit losing the target will relock quickly.

The proposed coordination policy is essentially a rule-based policy. It is easy to understand and implement with high operation stability, and it also can be improved combining with new scheduling algorithms [45,46]. It should be pointed out that this system can only coordinate and track a single target at present. When it is extended to multi-target tracking in the future, coordination and communication efficiency among tracking units need to be improved, and individually inferred communication [47] will be potential supporting technology.

## 4. Experiment Results

### 4.1. Target-Detection Experiments

To verify the effectiveness of our target-detection algorithm, a video from the East China Sea is selected as a test sample. In the experiment, TFD, MGBM, and our improved algorithm were used to detect a moving target, respectively. The processing results of frame 100, frame 200, and frame 500 using different methods were compared as shown in Figure 8.

The results using TFD are listed in Figure 8b. We can see that the contour of the moving target is extracted, and the contour of the dynamically changing background, namely, water ripple and coast, is introduced, which will cause a part of the background to be mistakenly included in the range of the moving target in subsequent extraction of the moving target. As shown in Figure 8c, when using MGBM only, the main part of the moving target in the image is detected. However, as the result of frame 500 shows, there is a hollow area in the extracted target. As the target extracted by MGBM is incomplete, only a part of the moving target can be extracted in subsequent processing of the moving-target contour. The improved algorithm can effectively solve the above problems. Figure 8d shows that the moving-target contour detected by our algorithm is more complete, and the position of the target is more accurate. Furthermore, the improved algorithm can adapt to a dynamic background, detect moving-target contour correctly, and extract moving-target effectively.

In addition, the average IoU, average recognition rate, and average false-detection rate obtained by the three algorithms are statistically analyzed for the whole image sequence. The statistical results are shown in Table 2.

As can be seen from Table 2, performance of the improved algorithm has been greatly improved in terms of average IoU, average recognition rate, and average false-detection rate. The average IoU of the proposed algorithm reaches 67.52%, which is much higher than that of MGBM and TFD. The average recognition rate of the proposed algorithm is 78.61%, while average false-detection rate is only 7.32%. Therefore, the improved moving-target-detection algorithm can effectively complete target-detection tasks with a dynamic background, and has better accuracy and robustness.

### 4.2. Target-Tracking Experiments

The dataset used in this paper includes an open-source dataset, Marine Obstacle Detection Dataset (MODD), and the dataset collected by our team. Each frame of the objects in the dataset is relabeled. There are 5070 pictures in the dataset. The proportion of the training, validation, and test sets is 6:2:2.

To test performance, our visual target-tracking method based on MDNet is compared with the appearance-model-based method in [31] and color-feature-based method in [33]. Figure 9 shows tracking results of frame 100, 300, 400, and 500 from MODD with pixels 480 × 640, in which the ship was driven to camera straightly and steadily. The red box in the image is the estimated boundary of the target using different methods, and blue rectangles represent the ground truth boundary box.

As shown in Figure 9, as the moving target in the video gradually approaches the camera, the target continues to grow in the picture. When using a tracking method based on the appearance model, the estimated target boundary box cannot position the target accurately, and will lose more and more features of the target. Similarly, results are not satisfactory either when using a color-feature-based method. At frame 500, the range of the estimated target frame is less than 1/3 of the actual size of the target. However, in testing of the method based on MDNet, the estimated target boundary box does not have a large deviation with longer tracking time, and the estimated target boundary box can completely include the moving target and accurately locate the position of the moving target.

Figure 10 shows the tracking results of frame 250, 450, 600, 650, 700, and 800 from a video of a fast-moving ship with pixels 480 × 700. Figure 10c shows that the estimated target boundary box does not drift significantly with tracking time using our method, and the obtained target position is still consistent with the actual position of the target when the ship turns at frames 650 and 700. Experimental results show that, when using our method based on MDNet, tracking error is smaller, and stability and robustness are significantly higher than that of compared methods, even for a fast-rotation target.

Location error and overlap (Intersection over Union, IoU) are used as analysis indexes to evaluate the performance of each method. Overlap rate represents the IoU of the estimated boundary box and ground truth boundary box, and location error represents Euclidean distance between the center position of the two boxes. If the frame overlap is greater and location error is smaller than corresponding thresholds, it is determined that prediction is accurate. The success rate is the ratio of video frames in which target is accurately tracked to all video frames. The results obtained from the test on the dataset are visualized as shown in Figure 11. It can be seen that the MDNet-based target-tracking method in this study has greater advantages in accuracy and success rate than the appearance-model-based and color-feature-based methods.

### 4.3. Target-Matching Experiments

To verify the proposed improved target-matching method, we use images of a USV in the same scene obtained by different cameras from different view angles, as well as images of three different USVs as input test samples. The SURF algorithm and our method combining SURF and RANSAC are used to match the targets.

Figure 12 shows matching results for different targets. Figure 12a shows there are 11, 13, and 12 pairs of feature points for three groups of USVs, respectively, using only the SURF algorithm. Since results are for different targets, all these are wrong-matched pairs of points. When using our method, all the mismatched pairs of points in Group 1 and Group 3 are screened out, and mismatched pairs of points in Group 2 are decreased to two, as shown in Figure 12b. The results show that our method can effectively decrease mismatching of moving targets.

We also have tests for the same moving target captured by different cameras from different view angles as shown in Figure 13. Figure 14 shows the corresponding data statistics of six groups of results in Figure 13.

From Figure 13 and Figure 14, it can be seen that, when matching the same moving target from different perspectives, if only SURF is used, the matching result contains a large number of false matching pairs of points, and the ratio of false matching points exceeds 40%, even up to 74.4%. The ratio of mismatched pairs of points decreased greatly when the mismatched points were screened with the RANSAC algorithm. It can be seen from the statistical line chart in Figure 13 that the trend of mismatching point-pairs rate remains below 20% on the whole. Experimental results show that the proposed target-matching algorithm combining the SURF algorithm and RANSAC algorithm has a better matching accuracy when it is used to deal with the same moving target under different viewing angles.

### 4.4. Field Experiments

In this paper, MTVT experiments were carried out in an area of Bohai Sea near Jinzhou City, Liaoning Province, China. There is a harbor enclosed by breakwaters and wave walls. The USV cruises into sea out of the harbor. We install tracking units on the top of the wave walls in north of the harbor with equal distance of 100 m, and USV with size of 6.5 × 2.0 × 1.5 m is taken as the target for tracking, as shown in Figure 15. Our system in the experiments is composed of three tracking units. Each unit consists of a Hikvision DS-2DC4223IW series PTZ camera, and an industrial control computer RK3288 with Ubuntu operation system and WIFI as shown in Figure 15c, tested Wi-Fi communication bandwidth is 90 Mbps, and average delay is 150 ms in our experiment condition. The resolution of each camera is set to 1280 × 720, and the image-acquisition frame rate is 25 fps. In the experiments, the USV cruises along the predefined path through its control system at a speed of around 10 knots.

In the first experiment, the initial optimal axes of the cameras are parallel with axis Y before the system begins tracking, as shown in the right of Figure 15. When the tracking system is started, each tracking unit detects the target USV from the environment automatically. Since the target is in the camera’s view field of Unit 3, it found the target first. Unit 3 acts as the MTU and turned to tracking state, as shown in the first row of Figure 16. At the same time, Unit 1 and Unit 2 rotate the PT to right and detect the moving target.

After 2 s, Unit 3 still acts as the MTU and tracks the moving target normally. At this time, Unit 1 and Unit 2 successfully complete detection and matching of the moving target and enter tracking state. The acquired image is shown in the second row of Figure 16. The three units dynamically adjust rotation direction of the camera PT according to the position of the target, aiming to adjust the target to the center of the camera’s field of view. By dynamically adjusting the angle, after 10 s, the three cameras all locate the target near to the center of the field of view, and realize smooth tracking of the moving target. The acquired images are shown in the third row of Figure 16.

During the experiment, angles of cameras’ optical axes are dynamically adjusted. Figure 17 depicts the process of camera angle change (clockwise is positive, counterclockwise is negative, and positive direction of the Y axis in Figure 15 is 0°).

As can be seen from Figure 17, after Unit 3 detects the moving target, it quickly adjusts angle to lock the moving target. Starting from 3 s, camera angle fluctuation gradually decreases, and the direction trend coincides with actual movement of the target, indicating that Unit 3 has implemented smooth tracking of the target. Similarly, cameras’ view angles of Unit 2 and Unit 1 are adjusted continuously and automatically, and changes of camera angles are consistent with the movement trend of the target.

In the second experiment, we test coordination performance of our system in a situation of target loss by part of the units. Target loss by part of the units easily happens in many scenarios. In this experiment, after the target sailed for 5 s, in the state shown in row 1 of Figure 18, we control Unit 3 to rotate about 50 degrees manually to make the target disappear in its field of view, while Unit 1 and Unit 2 track the moving target normally. At this time, Unit 3 turns into “out of lock” state, and images acquired by each camera are shown in row 2 of Figure 18. Unit 3 stops rotating for about 1 s, searches for moving targets in the field of view, and then turns to detection state, and patrols and looks for moving targets according to shared target information. After less than 3 s, as shown in row 3 of Figure 18, Unit 3 successfully retrieved tracking the target.

Table 3 shows, from 5 to 9 s, the number of frames that the target was tracked by three units during the process of retrieving the target. Unit 2 tracked the target in all frames of the 4 s. There are six frames where Unit 1 failed to lock target, but the unit retrieved target by itself in several frames. For Unit 3, because view angle was changed manually, in the process of PT rotation, there are 19 frames of image in which target the appeared, and target was detected, matched, and tracked in about 0.3 s. From Table 3, it can be seen that our MTVT system with multi-camera cooperation can keep tracking and help unit loss of targets to retrieve the target quickly.

In the third field experiment, we test the system when units have random initial view angles (as shown in Figure 19) and target moving speed changed. At the beginning of experiment, because each camera fails to detect a moving target, it rotates horizontally to search for the target. Unit 2 locks the target first in 2 s and then enters tracking state, acting as the MTU. Then, according to feedback results of Unit 2 and coordination strategies of the MTVT system, Units 1 and 3 rotate PT towards the direction of the target rapidly, and lock the target for tracking in 3 s, as shown in Figure 20. The view-angle adjusting process of the three units is shown in Figure 21a, and the speed of the target in the experiment is shown in Figure 21b. It can be seen that, when Unit 2 detected a moving target, the system could quickly respond and regulate other units to turn to the target to be tracked, and the system could still have good tracking results for a target whose moving speed changed.

To sum up, the MTVT system with multi-camera cooperation proposed in this paper can coordinate all tracking units to jointly perform tracking tasks when some units detect a moving target, and has high stability and continuity in the tracking process. When any unit loses a target, it can quickly retrieve the moving target according to tracking status of other units and shared target information. Compared with a tracking system with a single camera, it further expands monitoring range and effectively improves reliability, accuracy, and stability of visual tracking.

## 5. Conclusions and Prospects

Due to highly dynamic conditions, it is a challenge to track sea-surface visual targets. In this paper, a visual detection and tracking system for sea-surface moving targets based on multi-camera cooperation was proposed and designed, and key technologies including moving-target visual detection, tracking, matching among multiple cameras, and coordination strategies are studied, respectively. Experiments show that the study of multi-camera cooperative visual tracking for sea-surface targets in this paper not only expands monitoring range, but also realizes continuous tracking of targets among cameras, effectively improves the reliability and continuity of visual tracking, and achieves expected results.

Although comprehensive research about the sea-surface target tracking system was reported in this paper, there are still some aspects that need further study: (1) In the experiments, we test the system with three units and obtain expected results, however there is room for improvement in information interaction and communication efficiency, especially when the number of tracking units increases a lot in the system. (2) Due to the limitation of our test condition, we did not test the system when the tracking units were installed on mobile platforms such as USVs, so the performances of the system in a bumped and tossed environment should be further studied. Furthermore, it will also be important research work to use technology of transfer learning [48,49] to improve generalization ability of the system for applications in various environments. (3) The technologies of multi-camera cooperative tracking in this paper are researched for single-target tracking, but for multiple-target tracking problems in a complex scene, novel approaches are required.

It is believed that, in the near future, USVs will gradually become the backbone of maritime operations. They can carry a variety of equipment and perform a variety of military and civilian missions in sea for long time, such as maritime search-and-rescue and navigation, reconnaissance and patrol, counter-terrorism, and so on. The multi-camera cooperative tracking system proposed in this paper has good portability, and is expected to be deployed on mobile platforms such as USVs, so as to realize the multi-USV cooperative all-round sustainable tracking of moving targets on the sea surface.

## Figures and Tables

**Figure 1 sensors-22-00693-f001:**
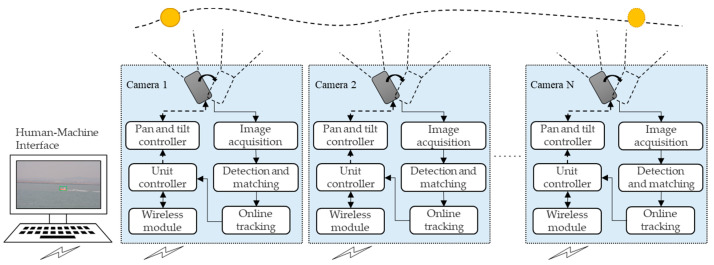
Diagram of the MTVT system with multi-camera cooperation.

**Figure 2 sensors-22-00693-f002:**
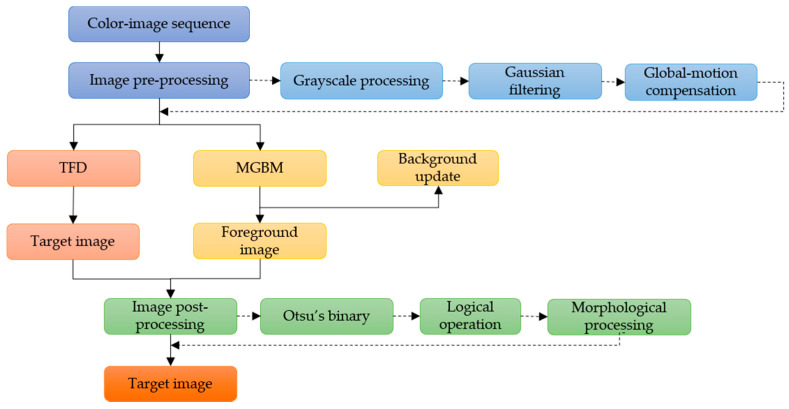
Flowchart of the improved moving-target visual-detection algorithm.

**Figure 3 sensors-22-00693-f003:**
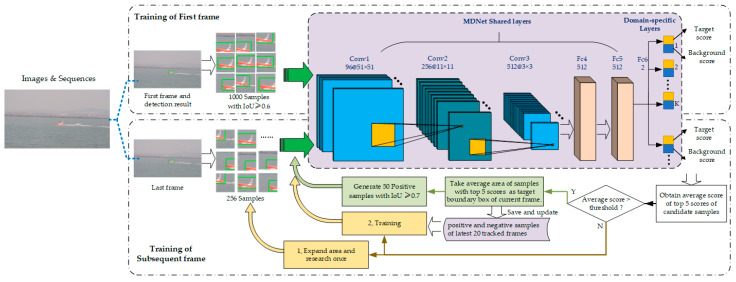
Construction and pretraining of MDNet.

**Figure 4 sensors-22-00693-f004:**
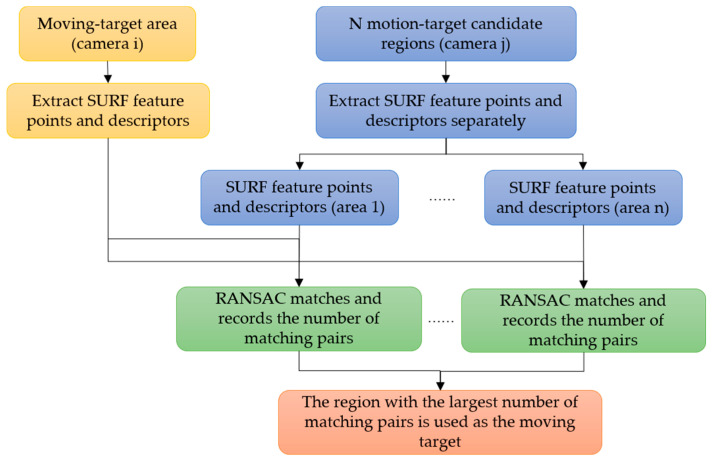
Flowchart of the improved moving-target-matching method.

**Figure 5 sensors-22-00693-f005:**
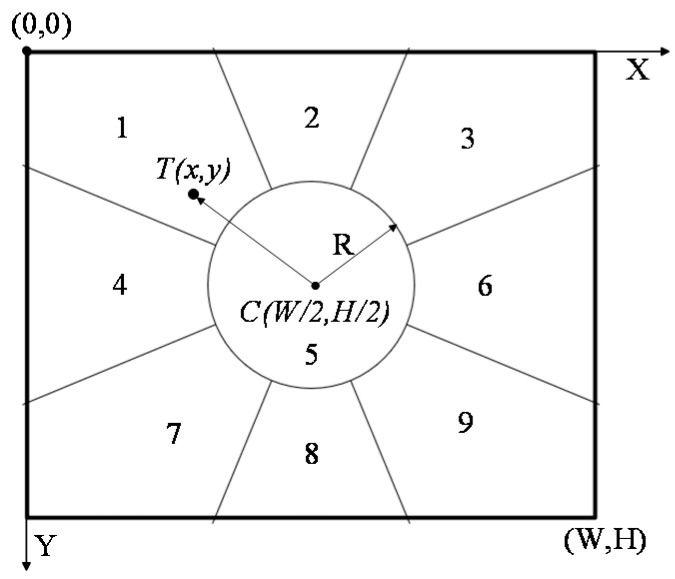
Region partition of the view field of camera.

**Figure 6 sensors-22-00693-f006:**
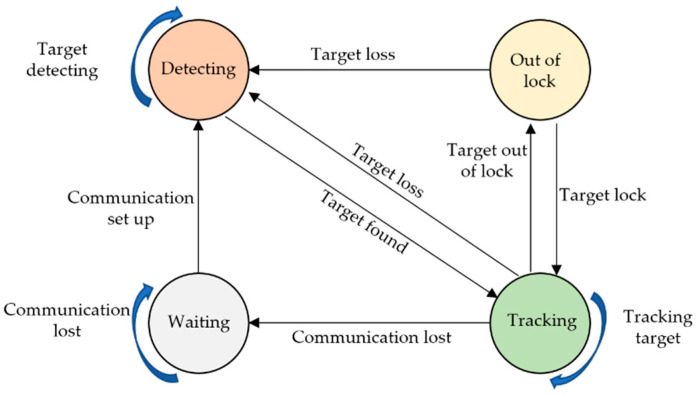
Schematic diagram of unit state transition.

**Figure 7 sensors-22-00693-f007:**
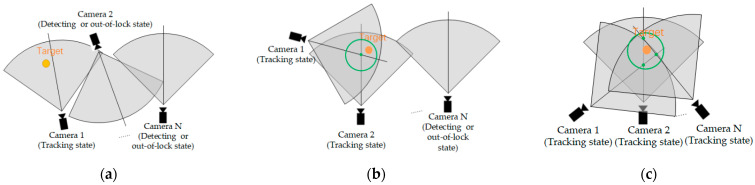
Pose relationship of cameras. (**a**) Pose relation of tracking beginning; (**b**) Pose relation of partial cooperation; (**c**) Pose relation of full cooperation.

**Figure 8 sensors-22-00693-f008:**
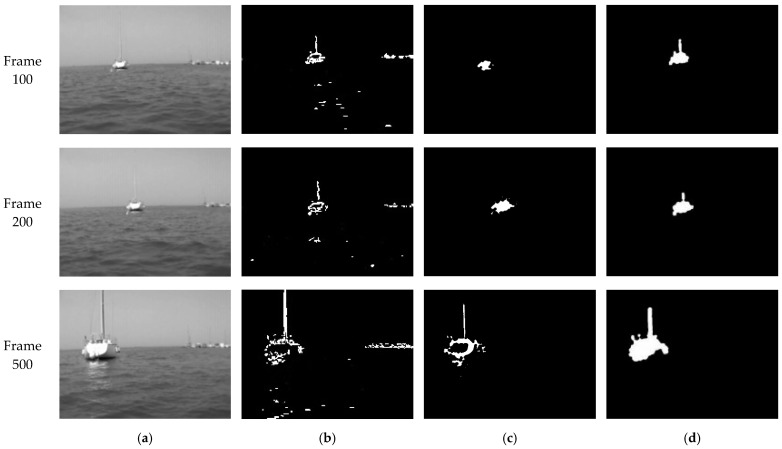
The target detection results: (**a**) Grayscale image of the original video image; (**b**) Result of TFD; (**c**) Result of MGBM; (**d**) Result of the improved algorithm.

**Figure 9 sensors-22-00693-f009:**
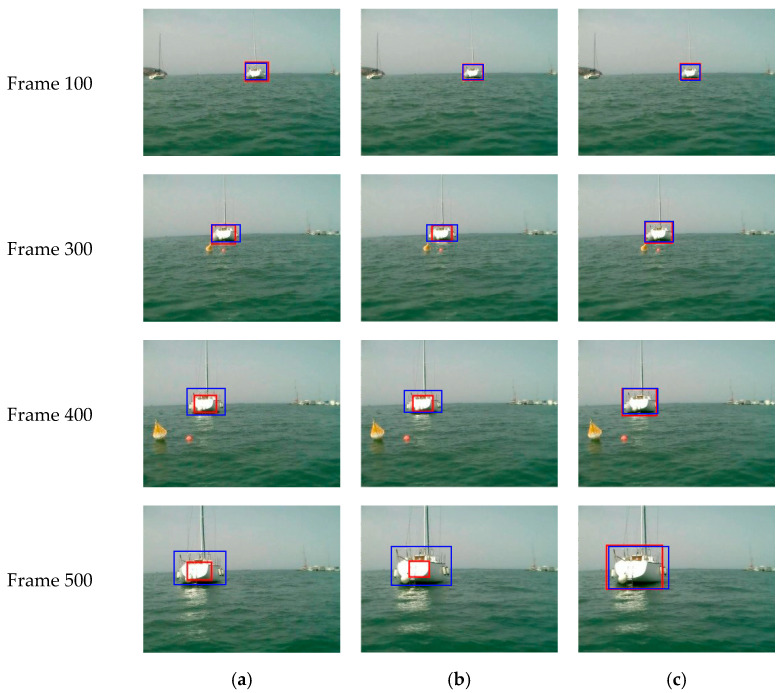
Visual target-tracking results of the sea-surface moving target. (**a**) Results of the appearance-model-based method. (**b**) Results of the color-feature-based method. (**c**) Results of our method.

**Figure 10 sensors-22-00693-f010:**
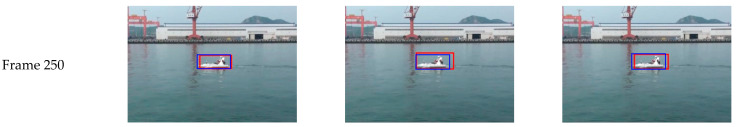
Visual target-tracking results of a sea-surface moving target with horizonal rotation motion. (**a**) Results of the appearance-model-based method. (**b**) Results of the color-feature-based method. (**c**) Results of our method.

**Figure 11 sensors-22-00693-f011:**
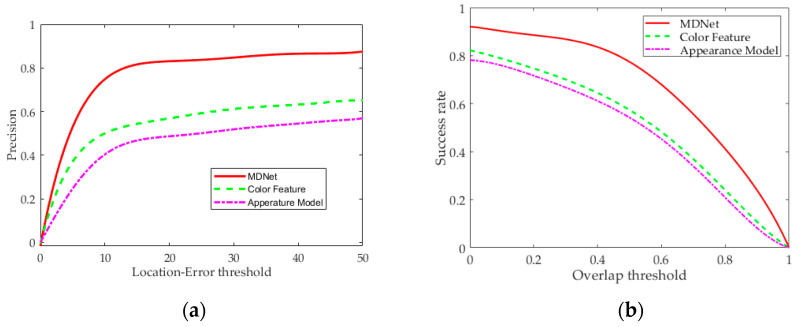
Test results of accuracy and success rate on datasets. (**a**) Accuracy curve. (**b**) Success-rate curve.

**Figure 12 sensors-22-00693-f012:**
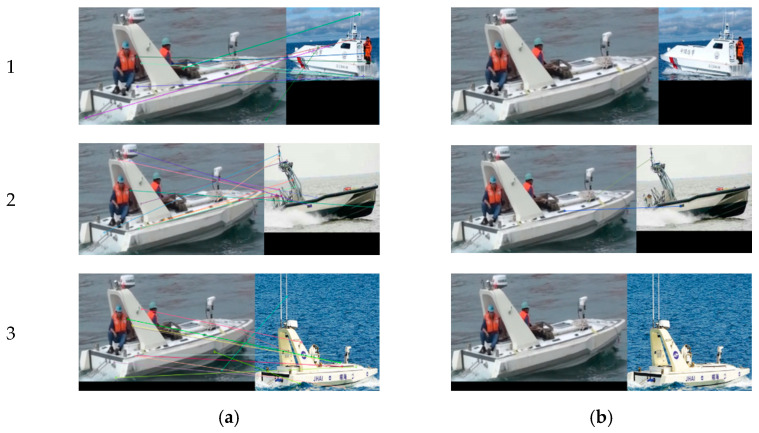
Matching results of different moving targets: (**a**) Results of the SURF algorithm; (**b**) Results of our method.

**Figure 13 sensors-22-00693-f013:**
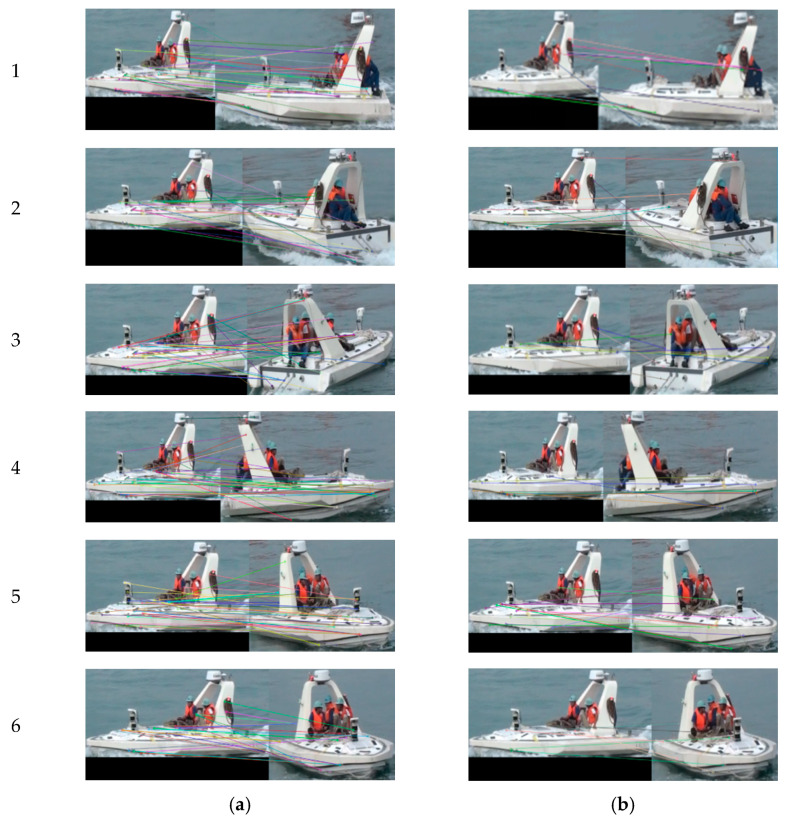
Matching results of the same moving target under different viewing angles: (**a**) Results of the SURF algorithm; (**b**) Results of our method.

**Figure 14 sensors-22-00693-f014:**
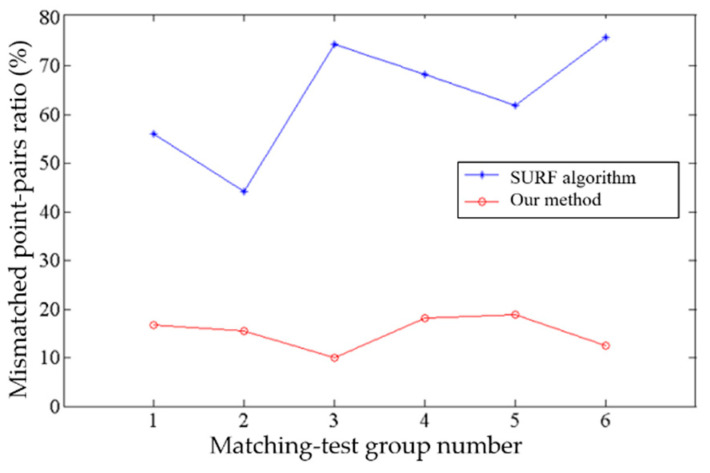
Ratio of mismatched pairs to all pairs.

**Figure 15 sensors-22-00693-f015:**
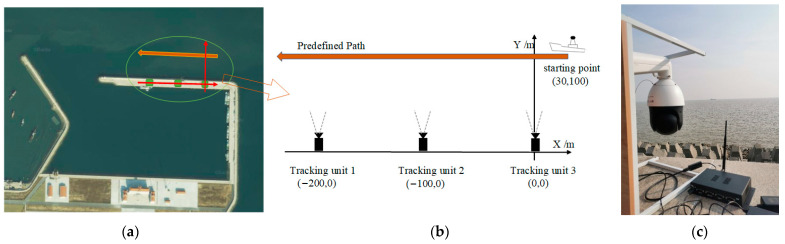
Experiment location and the position of tracking units and target. (**a**) Satellite map of experimental site. (**b**) The predefined path of the target, and the positions and initial view angles of three units. (**c**) Tracking-unit deployment scenario.

**Figure 16 sensors-22-00693-f016:**

Images and tracking results of three units: (**a**) Image of Unit 1; (**b**) Image of Unit 2; (**c**) Image of Unit 3.

**Figure 17 sensors-22-00693-f017:**
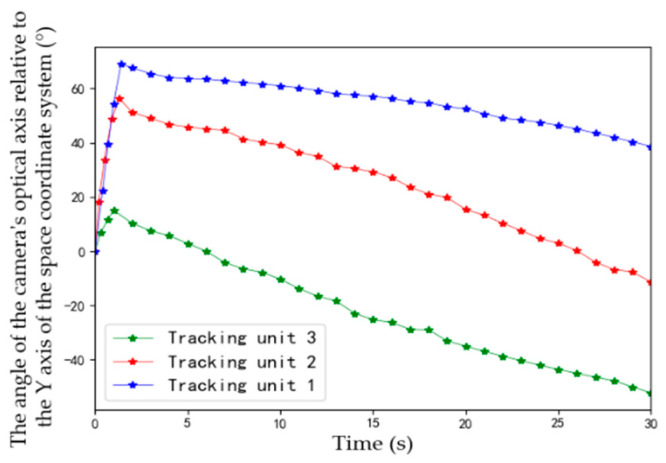
View-angle adjustment process of units.

**Figure 18 sensors-22-00693-f018:**
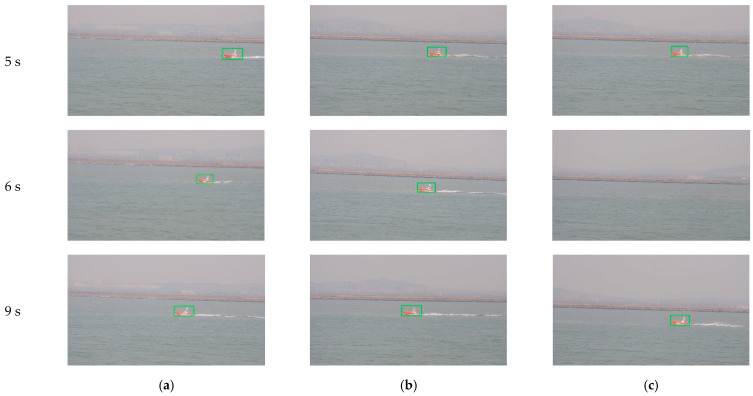
Images and tracking results of three units with a manual interruption: (**a**) Image of Unit 1; (**b**) Image of Unit 2; (**c**) Image of Unit 3.

**Figure 19 sensors-22-00693-f019:**
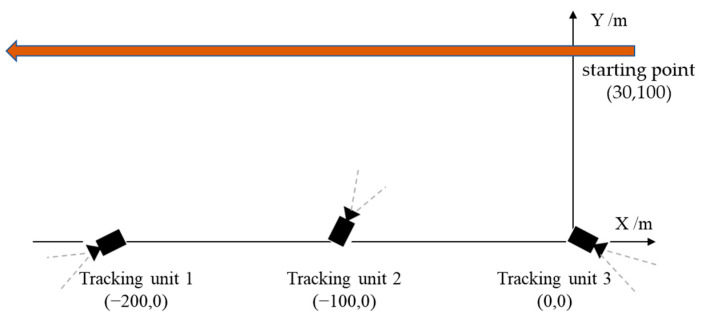
The position and initial view angles of three units.

**Figure 20 sensors-22-00693-f020:**
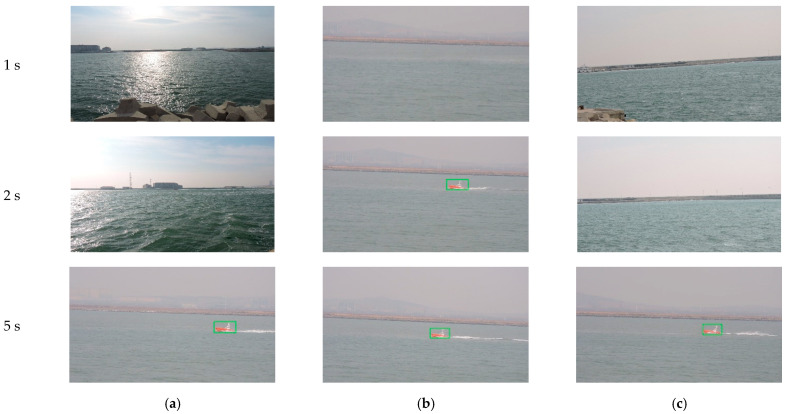
Images and tracking results of three units with random initial view angles: (**a**) Image of Unit 1; (**b**) Image of Unit 2; (**c**) Image of Unit 3.

**Figure 21 sensors-22-00693-f021:**
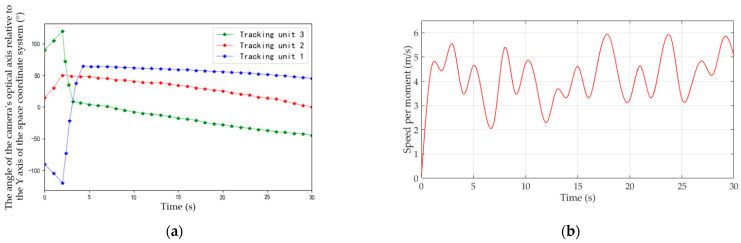
Results of the third field experiments: (**a**) The velocity of target; (**b**) View-angle adjustment process of units.

**Table 1 sensors-22-00693-t001:** Pan-and-tilt direction of camera.

Target Area	Range of Target Location	Pan-and-Tilt Direction
1	|***r***| ≥ *R*,$\vec{r}\in$ (9π/8, 11π/8]	Upper left
2	|***r***| ≥ *R*,$\vec{r}\in$ (11π/8, 13π/8]	Upper
3	|***r***| ≥ *R*,$\vec{r}\in$ (13π/8, 15π/8]	Upper right
4	|***r***| ≥ *R*,$\vec{r}\in$ (7π/8, 9π/8]	Left
5	|***r***| < *R*	Stop
6	|***r***| ≥ *R*,$\vec{r}\in$ (15π/8, 2π)∪[0, π/8]	Right
7	|***r***| ≥ *R*,$\vec{r}\in$ (5π/8, 7π/8]	Lower left
8	|***r***| ≥ *R*,$\vec{r}\in$ (3π/8, 5π/8]	Lower
9	|***r***| ≥ *R*,$\vec{r}\in$ (π/8, 3π/8]	Lower right

**Table 2 sensors-22-00693-t002:** Detection statistical data of the three algorithms.

Algorithms	Average IoU (%)	Average Recognition Rate (%)	Average False-Detection Rate (%)
TFD	32.68	54.17	23.49
MGBM	47.19	62.35	15.29
Our algorithm	67.52	78.61	7.32

**Table 3 sensors-22-00693-t003:** The number of frames of the target in each unit.

	Unit 1	Unit 2	Unit 3
The number of frames correctly tracked	94	100	11
The number of frames in which the target appears	100	100	19

## Data Availability

The data presented in this study are available on request from the corresponding author. The data are not publicly available due to the data also forms part of an ongoing study.

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
