# Peer review of "Sea-Surface Target Visual Tracking with a Multi-Camera Cooperation Approach"

_sensors, 2022, doi:10.3390/s22020693_

Round 1

Reviewer 1 Report

Summary of the paper

This paper proposes a new method for multi-camera sea surface target visual tracking. The method incorporates visual detection, tracking, matching, and multi-camera cooperation, forming a multi-camera visual target tracking pipeline. The visual detection module is an improvement of MGBM and TFD, while the tracking part is based on MDNet. SURF and RANSAC are combined to match the feature points of moving targets detected by different cameras. A rule-based camera control strategy is designed to enhance multi-camera cooperation. The experimental results show the advantage of the proposed method and verify the effect of each component.

Strengths:

The paper is well written and easy to follow. The proposed system for sea surface target tracking is complete and technically sound. The usability of the system is validated in real-world scenarios.

Weakness:

One of the weak points of this paper is that it seems to lack some novelty. The modules in visual detection, tracking, and matching are all based on existing methods. For example, the target tracking part is just an implementation of MDNet. Besides, I wonder why the authors do not adopt stronger tracking algorithms (e.g., SiameseRPN[1], ATOM[2], DiMP[3]), as the vision backbone?

The coordination of multi-camera is reached according to a rule-based control policy. There are a number of methods [4, 5] are proposed for this problem. So it is necessary to discuss the advantage and disadvantages of the proposed methods, comparing these existing solutions. Besides, is it possible to extend the designed method for multi-target tracking [6]?

Cameras rely on WIFI modules to communicate with each other and the HMI so as to exchange information necessary for cooperation. Therefore, communication efficiency is crucial for the application. So, it is necessary to test/analyze the communication bandwidth and delay in the experiment, referring to I2C[7] and ToM2C[8], to make the feasibility of the method be more convincing.

The effectiveness of vision modules are evaluated in a video dataset, while the cooperation strategy is tested in reality (Bohai Sea). Although the experiments in reality is interesting and important, I suggest the authors further validate the policy in diverse environments, e.g., different weather situation and different camera settings, to analysis the robustness of the system. Of course, it is also a feasible choice to simulate these situations in virtual environments.

Reference:

[1] Li, Bo, et al. "High performance visual tracking with siamese region proposal network." Proceedings of the IEEE conference on computer vision and pattern recognition. 2018.

[2] Danelljan, Martin, et al. "Atom: Accurate tracking by overlap maximization." Proceedings of the IEEE/CVF Conference on Computer Vision and Pattern Recognition. 2019.

[3] Bhat, Goutam, et al. "Learning discriminative model prediction for tracking." Proceedings of the IEEE/CVF International Conference on Computer Vision. 2019.

[4] Li, Jing, Jing Xu, Fangwei Zhong, Xiangyu Kong, Yu Qiao, and Yizhou Wang. Pose-assisted multi-camera collaboration for active object tracking. In Proceedings of the AAAI Conference on Artificial Intelligence, vol. 34, no. 01, pp. 759-766. 2020.

[5] Kumari, Pratibha, et al. "Dynamic Scheduling of an Autonomous PTZ Camera for Effective Surveillance." 2020 IEEE 17th International Conference on Mobile Ad Hoc and Sensor Systems (MASS). IEEE, 2020.

[6]Ding, Ziluo, Tiejun Huang, and Zongqing Lu. "Learning individually inferred communication for multi-agent cooperation." arXiv preprint arXiv:2006.06455 (2020).

Reviewer 2 Report

  1. The size of figure 4 is small, please enlarge it for better view.
  2. Due to the limitation of testing condition, have you considered to use transfer learning technique to improve the generalization ability? such as [1] J. Zheng, et. al., "Improving the Generalization Ability of Deep Neural Networks for Cross-Domain Visual Recognition", IEEE Transactions on Cognitive and Developmental Systems, vol. 13, no. 3, pp. 607-620, 2021. [2] C. Hao, et. al., "Software/Hardware Co-design for Multi-modal Multi-task Learning in Autonomous Systems",  IEEE 3rd International Conference on Artificial Intelligence Circuits and Systems (AICAS), 2021.
  3. Please provide details of your dataset, such as the number of training, validation, and test sets.

Round 2

Reviewer 1 Report

I have read the author's response and the revised version. Most concerns are addressed in the response, so I think it is ready to be accepted. Notably, the format of figures should be vector diagram.